# Bile Acid Receptors and the Gut–Liver Axis in Nonalcoholic Fatty Liver Disease

**DOI:** 10.3390/cells10112806

**Published:** 2021-10-20

**Authors:** Rui Xue, Lianyong Su, Shengyi Lai, Yanyan Wang, Derrick Zhao, Jiangao Fan, Weidong Chen, Phillip B. Hylemon, Huiping Zhou

**Affiliations:** 1Department of Gastroenterology, Shanghai Key Lab of Pediatric Gastroenterology and Nutrition, Shanghai 210092, China; jasminxue@sjtu.edu.cn (R.X.); fanjiangao@xinhuamed.com.cn (J.F.); 2Department of Microbiology and Immunology, Medical College of Virginia and McGuire Veterans Affairs Medical Center, Virginia Commonwealth University, Richmond, VA 23284, USA; Liangyong.su@vcuhealth.org (L.S.); slai2@alumni.vcu.edu (S.L.); Derric.Zhao@vcuhealth.org (D.Z.); Phillip.hylemon@vcuhealth.org (P.B.H.); 3School of Pharmaceutical Science, Anhui University of Chinese Medicine, Hefei 230031, China; wangyanyan@ahtcm.edu.cn (Y.W.); wdchen@ahtcm.edu.cn (W.C.)

**Keywords:** bile acids, gut–liver axis, bile acid receptor, FXR, TGR5, S1PR2, NAFLD

## Abstract

The prevalence of nonalcoholic fatty liver disease (NAFLD) has been significantly increased due to the global epidemic of obesity. The disease progression from simple steatosis (NAFL) to nonalcoholic steatohepatitis (NASH) is closely linked to inflammation, insulin resistance, and dysbiosis. Although extensive efforts have been aimed at elucidating the pathological mechanisms of NAFLD disease progression, current understanding remains incomplete, and no effective therapy is available. Bile acids (BAs) are not only important physiological detergents for the absorption of lipid-soluble nutrients in the intestine but also metabolic regulators. During the last two decades, BAs have been identified as important signaling molecules involved in lipid, glucose, and energy metabolism. Dysregulation of BA homeostasis has been associated with NAFLD disease severity. Identification of nuclear receptors and G-protein-coupled receptors activated by different BAs not only significantly expanded the current understanding of NAFLD/NASH disease progression but also provided the opportunity to develop potential therapeutics for NAFLD/NASH. In this review, we will summarize the recent studies with a focus on BA-mediated signaling pathways in NAFLD/NASH. Furthermore, the therapeutic implications of targeting BA-mediated signaling pathways for NAFLD will also be discussed.

## 1. Introduction

Nonalcoholic fatty liver disease (NAFLD) is the most common chronic liver disease worldwide and is rapidly emerging as the leading cause of end-stage liver disease due to the global epidemic of obesity, type 2 diabetes, and metabolic syndrome [1]. It is well accepted that the disease progression from simple steatosis with little or no inflammation (nonalcoholic fatty liver (NAFL)) to nonalcoholic steatohepatitis (NASH), fibrosis, cirrhosis, and hepatocellular carcinoma (HCC) is promoted by multiple factors and is closely associated with sex, age, and metabolic status [2,3,4,5,6,7,8]. The disruption of intrahepatic bile acid (BA) homeostasis, aberrant activation of the innate immune response, insulin resistance, and dysbiosis are major driving forces of the disease progression from NAFL to NASH, cirrhosis, and HCC [9,10,11,12]. It has been well established that NAFLD is a multisystem disease and is associated with cardiovascular diseases, kidney diseases, and gastrointestinal diseases. Furthermore, NAFLD is a major risk factor for both hepatic and extrahepatic malignancies such as colon cancer and pancreatic cancer [13,14]. Even though extensive studies have been performed during the last two decades, no effective therapy for NAFLD has been developed due to the limited understanding of the pathogenesis of this complex disease.

BAs are exclusively synthesized from cholesterol in the hepatocytes. BAs are the key organic components of bile stimulating bile flow [15]. Intrahepatic BA circulation is an important physiological process. BAs play an essential role in emulsifying and absorbing dietary fat, cholesterol, and fat-soluble vitamins. Since the discovery of the first BA nuclear receptor, farnesoid receptor X (FXR), in 1999, BAs have been extensively studied for their role as critical signaling molecules and for their involvement in various physiological and pathological processes [16]. Identification of BA-activated G-protein-coupled receptors (GPCRs), Takeda GPCR 5 (TGR5, also known as GPBAR1) and sphingosine-1 phosphate receptor 2 (S1PR2), has made a significant advance by linking BAs with various metabolic diseases, including NAFLD [17,18,19].

The gut microbiome is composed of a diverse range of microbes. These microbes play critical roles in maintaining intestinal barrier function, modulating metabolic processes and immune responses [20]. One major function of gut microbes is the biotransformation of BAs. Dysbiosis disrupts not only intestinal barrier function but also has detrimental effects on the liver [21,22,23]. As important messengers of communication between the gut and the liver, BAs and their metabolites have been well recognized for their influence on lipid and energy metabolism, as well as their impact on immune responses [10,12,24]. Therefore, disruption of BA homeostasis can have pathological consequences to the liver and the gut. In this review, we will focus on the role of BAs and their receptors in the gut–liver axis in the disease progression of NAFLD. We will also discuss the therapeutic implications of targeting BA signaling pathways for NAFLD and NASH.

## 2. Enterohepatic Circulation and BA Metabolism

Enterohepatic circulation is an essential physiological process that circulates BAs from the liver to the bile, followed by entry into the intestinal lumen, absorption by the enterocytes in the ileum, and transport back to the liver via the portal vein (Figure 1). The efficiency of the enterohepatic cycling of BAs is exceptionally high. About 95% of BAs are recovered from the gut during each enterohepatic circulation. In the small intestine, BAs are absorbed by passive absorption. In the terminal ileum, BAs are absorbed by active transport. Only 5% (400–800 mg/d) of BAs are excreted via feces and replaced by *de novo* synthesis from cholesterol in the liver. The human BA pool size is about 4–6 g. The daily amount of BAs secreted into the intestine is approximately 12–30 g in adults. The average number of daily BA recycling is between 3 and 5 times [25].

### 2.1. Bile Acid Synthesis

BAs are exclusively synthesized from cholesterol in the hepatocytes as the dominant catabolic pathway of cholesterol. The two major biosynthetic pathways of BAs have been well characterized, known as the classical pathway and alternative pathway [26]. The classical pathway is also called the “neutral” pathway since neutral intermediate metabolites are formed during the process. It is responsible for the production of the majority of BAs (>90%) under normal physiological conditions. In the classical pathway, cholesterol is converted into two primary BAs, cholic acid (CA) and chenodeoxycholic acid (CDCA), in approximately equal amounts in humans. The process of conversion from cholesterol to BAs requires more than a dozen enzymes to modify the cholesterol steroid core, removing the sidechain and conjugating with taurine or glycine. Cholesterol-7α-hydroxylase (CYP7A1), a cytochrome P450 enzyme and exclusively expressed in the liver, is the rate-limiting enzyme in this pathway and catalyzes the first step in BA synthesis, 7α-hydroxylation. The resulting 7α-hydroxycholesterol can be further modified by a sidechain shortening and reduction of the steroid double bond, followed by sidechain oxidation by mitochondrial sterol 27-hydroxylase (CYP27A1) to form CA and CDCA. The alternative pathway is also known as the “acidic pathway” since acidic intermediate metabolites are formed. It is initiated by CYP27A1, the rate-limiting enzyme in this pathway, followed by 7α-hydroxylation by oxysterol 7α-hydroxylase (CYP7B1). This pathway is believed to form mostly CDCA. Our recent study reported that CYP7B1 controls the levels of intracellular regulatory oxysterols generated by the “acidic/alternative” pathway of cholesterol metabolism [27,28]. CYP7B1 is widely expressed and has been reported to regulate steroid hormone metabolism in the brain [29,30]. Genetic mutations of CYP7B1 have been linked to inborn errors of BA metabolism [31]. CYP27A1 is a mitochondrial enzyme and is also expressed in macrophages and other cells [32]. However, hepatocytes have all the enzymes required for BA synthesis. In humans, primary BAs are conjugated with glycine or taurine through the action of a BA: CoA synthetase (BACS) and BA-CoA: amino acid N-acyltransferase (BAAT). BAs also can be sulfated (sulpho-) or glucuronidated (glucuronic-) by sulfotransferases (SULTs) and UDP-glucuronosyltransferases (UGTs) to increase solubility, reduce toxicity, and facilitate urinary excretion [33]. The BA pool compositions in rodents are substantially different from humans. As shown in Figure 2, in mice, CDCA and UDCA are converted to α-muricholic acid (α-MCA) and β-MCA via 6β-hydroxylation by the cytochrome P450, Cyp2c70, respectively [34,35]. A recent study using Cyp2c70 knockout mice reported that in the absence of Cyp2c20, the accumulation of UDCA was much less than that of CDCA, suggesting that most of β-MCA is derived from CDCA via α-MCA [36]. In humans, most primary BAs are glycine conjugated, while in mice, the majority of primary BAs are taurine conjugated.

BA synthesis is tightly regulated. The major feedback suppression mechanism controlling BA homeostasis has been well characterized [37,38]. BA-induced activation of FXR induces the expression of the small heterodimer partner (SHP), which suppresses the transcription of CYP7A1 by inhibiting hepatic nuclear factor 4 and liver-related homolog 1 [37,38]. In addition, fibroblast growth factor 19 (FGF19)/FGF15 released from intestinal epithelial cells also plays a critical role in regulating BA synthesis by activating FGF receptor (FGFR) 4 (FGFR4) and SHP [39]. Furthermore, sexual dimorphism contributes to the gender difference in BA synthesis. The key enzymes involved in BA synthesis are regulated by estrogen, impacting BA concentration and composition [40,41]. Therefore, it is important to take the consideration of sex differences in the treatment of NAFLD.

### 2.2. Biotransformation of BAs

Once the BAs are secreted into the gastrointestinal tract, a small portion of BAs escape the reabsorption and will be deconjugated and modified by gut bacteria via various biotransformations, such as oxidation, esterification, epimerization of hydroxyl groups, and desulfation to generate a variety of secondary BAs. In humans, more than 50 secondary BAs have been identified in fecal samples [42]. The initial step of the secondary BA formation is deconjugation, which is mediated by bile salt hydrolase (BSH), which is present in many bacteria, including Gram-negative Bacteroides and Gram-positive Lactobacillus and Clostridium. The free BAs can either pass the gut barrier via passive diffusion or be further modified by bacteria. CA and CDCA are oxidized and followed by 7α-dehydroxylation to form the secondary BAs, deoxycholic acid (DCA) and lithocholic acid (LCA), respectively [43,44]. In addition, CDCA can be transformed into 3α,7β-dihydroxy-5β-cholanoic acid (UDCA) by 7α and 7β-hydroxysteroid dehydrogenases (HSDH) via C7α/β-epimerization [45]. Unlike oxidation and epimerization, the 7α-dehydroxylation is only processed by a small number of anaerobes in the gut belonging to the genus *Clostridium* [42]. In mice, Tα-MCA and Tβ-MCA are deconjugated by BSH to form α-MCA and β-MCA. α-MCA is further transformed into murideoxycholic acid (MDCA) and hyodeoxycholic acid (HDCA), and β-MCA is transformed into ω-MCA. Although MDCA and HDCA can be synthesized from LCA by Cyp3a, gut bacteria-mediated transformation of α-MCA is the major source of MDCA and HDCA [36] (Figure 2). These secondary BAs can also be passively absorbed from the gut and function as signaling molecules. A recent study reported that the secondary BAs could be converted back to primary BAs by Cyp2a12 in mice [36].

### 2.3. Bile Acid Transporters

BAs are amphipathic molecules containing both hydrophilic (α-hydroxyl group) and hydrophobic (β configuration of H on C5) faces [46] (Figure 3). The different BAs differ markedly in hydrophobic and hydrophilic characters, which account for the various physiological and biological functions of BAs. The enterohepatic circulation of BAs depends on the active transport systems in the liver and intestine. As illustrated in Figure 4, BAs in the hepatocytes, either from *de novo* synthesis or recycling from the gut, are exported out of cells by the bile salt export pump (BSEP, gene symbol ABCB11). BSEP is located at the canalicular membrane of hepatocytes and is responsible for the secretion of BAs into the canalicular lumen. The genetic mutations of ABCB11 are linked to progressive type II familial intrahepatic cholestasis [47]. The major transporters responsible for hepatic sinusoidal BA uptake are the Na^+^-taurocholate cotransporting polypeptide (NTCP; gene symbol SLC10A1) and Na^+^-independent organic anion-transporting polypeptide (OATP, gene symbol SLCO). Compared to NTCP, the driving force responsible for OATP-mediated uptake is not well understood. In the basolateral membrane, the multidrug resistance-associated protein 2 (MRP2, gene symbol ABCC2), MRP4 (gene symbol ABCC3), and organic solute transporter α/β (OSTα/β, gene symbols SLC51A and SLC51B) provide alternative excretion routes for BAs into the systemic circulation [33]. In the cholehepatic shunt, the apical sodium-dependent BA transporter (ASBT) in the apical membrane of cholangiocytes is responsible for BA uptake. MRP3 and OSTα/β at the basolateral surface are responsible for the excretion of BAs into the hepatic arterial circulation.

Most BAs in the intestine are reabsorbed at the terminal ileum, where the bile acid transporters are highly expressed. The ileal active transport system is the primary route for conjugated BA uptake, especially for the more hydrophilic and taurine-conjugated BAs. ASBT, also known as the ileal BA transport (IBAT), is able to cotransport Na^+^ with BA at the apical membrane. Once the BAs are uptaken by intestinal epithelial cells, their efflux is carried out by OSTα/β. In addition, it has been shown that MRP3 may also be involved in BA efflux, at least in rodents [25].

## 3. BAs as Signaling Molecules

BAs have long been known as detergents to facilitate lipid and nutrient absorption in the intestine. Since the identification in 1999 of the nuclear hormone receptor (NHR), FXR, as a BA-activated receptor, BAs have been extensively studied as critical signaling molecules regulating lipid, glucose, and energy metabolism [32,48,49,50,51]. The identification of BA-activated GPCRs, TGR5, and S1PR2 further expanded the role of BAs in regulating metabolic pathways [18,52,53,54].

### 3.1. Bile-Acid-Mediated Activation of FXR

FXR (gene symbol NR1H4) is expressed in the liver, intestine, colon, kidney, adrenal gland, and ovary. The highest expression levels of FXR are detected in the liver and gastrointestinal tract. FXR is optimally activated by unconjugated BAs and regulates BA synthesis in a tissue-specific manner [55]. In hepatocytes, FXR activation negatively regulates BA synthesis by inducing the expression of a small heterodimer partner (SHP, NR0B2). SHP is a suppressor of CYP7A1 expression, the rate-limiting enzyme in BA synthesis. The activity of FXR is enhanced by forming a heterodimer with the retinoid X receptor (RXR), which provides an additional level of regulation of FXR-mediated signaling pathways [56,57]. RXR is a common heterodimerization partner of nuclear receptors [57]. In the ileal part of the small intestine, activation of FXR induces expression of fibroblast growth factor 19 or 15 (FGF19 in humans and FGF15 in mice). FGF19/15-mediated suppression of CYP7A1 is mediated by FGFR4 and SHP [39] (Figure 5). BAs differ in their potency in activating FXR. CDCA is the highest, followed by DCA, LCA, and CA [48]. In mice, CDCA and UDCA are converted to α-MCA and β-MCA by Cyp2c70, respectively [35]. It has been identified that taurine-conjugated α-MCA and β-MCA are natural FXR antagonists [58]. The role of FXR in hepatic metabolism has been reviewed recently [48,55,59].

### 3.2. Bile-Acid-Mediated Activation of GPCRs

#### 3.2.1. TGR5

TGR5 is initially identified in macrophages as the first GPCR activated by BAs [53]. TGR5 is ubiquitously expressed in all tissues including in the gastrointestinal tract, liver/gallbladder, kidney, brown adipose tissue, lymphoid tissues, adipose, and lung [54]. In the liver, TGR5 is mainly expressed in nonparenchymal cells and absent in hepatocytes [60]. A recent study reported that TGR5 is expressed in hepatocytes using albumin-Cre-TGR5 knockout mice [17]. However, albumin is not hepatocyte specific. TGR5 has a higher affinity with secondary BAs than primary BAs. LCA is the potent natural agonist of TGR5, followed by DCA, CDCA, and CA. Taurine conjugation of LCA and DCA further enhanced their TGR5 agonistic activity, rendering TLCA and TDCA the most potent TGR5 agonists [52]. However, CA, TCA, and GCA are weak agonists of TGR5 (Figure 6). Since the identification of TGR5, it has been extensively studied as an important regulator for glucose and energy metabolism [60]. The roles of TGR5 in liver diseases have been extensively reviewed recently [61,62,63,64].

#### 3.2.2. S1PR2

S1PR2 is one of the five S1PRs, which was originally discovered as endothelial differentiation G-protein-coupled receptor 5 (EDG5) [65]. During the last two decades, extensive studies have been performed to identify the specific role of individual S1PRs under normal physiological and various disease conditions. The differential expression of each S1PR in different cells and tissues and their coupling to specific G proteins render their unique biological functions [66]. S1PR2 is widely expressed. It has been implicated in many physiological and pathological processes and plays critical roles in regulating the immune response, metabolic processes, and cardiovascular, renal, and musculoskeletal functions, as well as the nervous system [67]. S1PR2 is highly expressed in the gastrointestinal tract and the liver. Identification of S1PR2 as a BA-activated GPCR in hepatocytes opened up a new direction of BA research [18]. Unlike the TGR5, S1PR2 is expressed in all hepatic cells and is the predominant S1PR in hepatocytes. S1PR2 is only activated by conjugated primary BAs. TCA is the most potent agonist. TCA-mediated activation of S1PR2 is critical to hepatic lipid and glucose metabolism (Figure 7) [68]. A recent study reported that both FTP720 and JTE-013 induced inflammation and promoted liver fibrosis in a WD-fed melanocortin-4 receptor (Mc4r)-deficient NASH mouse model by inducing aberrant methylation [69]. However, this study mistakenly used FTP720 as an antagonist for S1PR1, S1PR4, S1PR5, and S1PR3. Indeed, FTP720 is a high-affinity agonist for S1PRs, except S1PR2 [70]. It has been reported that activation of SphK1/S1PRs signaling promotes proinflammatory and fibrotic responses and aggravates liver injury. In addition, its activity is dependent on the phosphorylation by SphK2 in vivo [71]. In this study, the authors also reported that the SphK2 expression was suppressed. It remains unclear whether the effect of FTY720 is S1PR-specific or not. Our previous study also reported that BA-induced activation of S1PR2 promotes cholangiocarcinoma cell proliferation and invasion [72]. More studies are needed in order to develop S1PR2-specific therapy for metabolic disease.

#### 3.2.3. Muscarinic Receptors

It has been reported that TLCA activates muscarinic receptor 3 (M3), but not M1 and M2 in gastric chief cells. The sulfated TLCA binds to M1, but not M2 and M3. TLCA-mediated activation of M3 inhibits acetylcholine-induced increases in inositol phosphate formation and activation of mitogen-activated protein kinase (MAPK). However, the concentration needed for TLCA to inhibit acetylcholine-mediated effects is at a mM range [73].

## 4. Gut–Liver Axis in NAFLD

The liver is the largest secretory gland in the human body and plays a crucial role in lipid, glucose, and energy metabolism. Due to the complexity and limited understanding of NAFLD disease pathogenesis, no regulatory-approved therapy is available, and liver transplantation remains the only therapeutic option for end-stage liver disease patients [74]. There is an unmet need to identify the underlying cellular and molecular mechanisms for developing effective therapies.

The maintenance of metabolic homeostasis both in the liver and gut is critical for many normal physiological processes. The gut and liver are constantly communicating with each other via numerous metabolites. BAs, as the crucial messengers, not only regulate hepatic metabolic functions but also impact the gut microbiome. Vice versa, the gut microbiome plays a critical role in maintaining intestinal homeostasis and exerts its influence on the liver by modifying BA composition, changing the BA pool, and regulating immune response [75,76,77]. Disruption of enterohepatic BA circulation is consequential to the liver and gut. The bidirectional influence of the gut microbiome and liver constitutes the gut–liver axis [78]. The impaired gut–liver axis has been implicated in various diseases, including NAFLD/NASH [78,79].

Recent studies also showed that the circulating BA profile is significantly changed and is correlated to NAFLD disease severity [80]. The ratio of conjugated primary BAs to unconjugated primary BAs is significantly increased in NASH patients but not in NAFL patients. Interestingly, total conjugated primary BAs and especially conjugated cholate (TCA + GCA) are significantly increased in NASH patients compared to NAFL [80]. A recent study with a large patient cohort further examined the correlations of plasma BA levels with NASH and insulin resistance. The results indicated that plasma concentrations are only elevated in NASH patients with pronounced insulin resistance [81]. Interestingly, this study also found that the plasma levels of conjugated primary BAs (TCA and GCA) efficiently discriminate NASH status independent of other confounding factors such as obesity and insulin resistance [81]. Our recent studies also showed that serum levels of TCA and TβMCA were markedly increased in high-fat diet and high-sugar water-induced NASH mouse models [82]. TCA-induced activation of S1PR2 is involved in cholangiocyte proliferation and hepatic fibrosis [83]. In addition, hepatic inflammation and gut barrier dysfunction are also associated with NASH disease progression [74] (Figure 8). Our previous study reported that activation of S1PR2 promotes intestinal epithelial cell proliferation [33].

The gender-specific differences in gut microbiota composition have been well established [84]. Changes in gut microbiome composition not only impact BA metabolism but also induce a different immune response. It is imperative to take into consideration biological sex differences in the development of therapeutic strategies to target the gut–liver axis for NAFLD and other metabolic diseases.

## 5. Targeting BA Receptors as Potential Therapeutics for NAFLD

GPCRs and NRs are major pharmacological targets [85]. The biosynthesis of the primary BAs in hepatocytes is tightly regulated through a classical negative feedback loop under normal physiological conditions [26,43]. Activation of FXR in hepatocytes and enterocytes results in the transduction of SHP and release of FGF19 (FGF15 in mice), respectively [86]. FGF19 further induces SHP expression and activation of ERK via the FGFR4/β-Klotho complex [55]. SHP is a transcription suppressor of CYP7A1. The hepatic BA composition and level have a direct impact on hepatic metabolism. As discussed in the previous sections, BAs differ in their affinity for different BA receptors. It has been well characterized that in hepatocytes, activation of FXR not only reduces BA synthesis but also increases BA secretion and conjugation. In addition, FXR activation inhibited glycolysis and lipogenesis via inhibiting carbohydrate-responsive element-binding protein (ChREBP) and sterol responsive element binding protein 1 (SREBP1c), respectively [87]. Systemic activation of FXR prevents hepatic lipid accumulation and reduces inflammation and fibrosis in the NASH mouse model [55]. It also has been reported that intestinal specific knockout of FXR or selectively inhibits intestinal FXR improves NAFLD and obesity-related metabolic dysfunction [88,89]. In contrast, it also has been reported that an intestinal-specific FXR agonist reduced body weight and lipogenic gene expression via inducing FGF15 and altering the gut microbiome, which led to alteration of BA composition, browning of adipose tissue, and increase in energy expenditure in a high fat diet-fed mouse model [90]. The intestinal-specific activation of FXR by an intestine-restricted FXR agonist fexaramine increased TLCA and LCA levels, which are strong endogenous agonists of TGR5. Activation of TGR5 stimulates GLP-1 secretion to improve insulin sensitivity and hepatic metabolism [90,91,92]. These studies indicate that FXR-gut microbiome-TGR5/GLP-1 signaling cascade represents a major mechanism underlying fexaramine-mediated beneficial effects on NAFLD and metabolic diseases. The role of TGR5 in glucose and energy metabolism has been well recognized and extensively reviewed [93,94,95,96,97,98]. The contribution of TGR5-mediated signaling in hepatic lipid metabolism has received less attention due to the absence of TGR5 in hepatocytes. However, TGR5 has been identified as a negative regulator of hepatic inflammation [18,64,99,100]. A dual FXR and TGR5 agonist, BAR502, has been reported to promote browning of white adipose tissue and reverse hepatic steatosis and fibrosis in HFD-fed and CCl4-induced mouse models [101]. Since obeticholic acid (OCA) was approved by FDA for the treatment of primary biliary cholangitis, research interests are significantly increased in the development of selective FXR modulators to reduce the potential side-effects of the conventional FXR agonist for NASH. A number of FXR agonists have been developed and reached phase I or phase II clinical trials for NASH [102,103,104]. The development of TGR5-targeted therapy has been compromised by various adverse effects of TGR5 agonists, such as gastrointestinal side effects, tumorigenesis, and systemic toxicity [62,95]. As shown in Figure 9, targeting intrahepatic BA circulation represents promising therapeutics for NASH. The major clinical trials targeting enterohepatic BA circulation for the treatment of NAFLD/NASH are listed in Table 1. 

Activation of S1PR2 by conjugated primary BAs plays an essential role in the maintenance of hepatic lipid metabolism. TCA/S1PR2-mediated activation of ERK and AKT signaling pathways and nuclear SphK2 is critical to maintaining hepatic lipid homeostasis. Activation of nuclear SphK2 has a significant impact on gene transcription by modulating histone acetylation. In S1PR2^−/−^ and SphK2^−/−^ mice, HFD-feeding rapidly induced hepatic lipid accumulation and dysregulation of BA metabolism [19,68]. In cholangiocytes, activation of S1PR2 promotes cell proliferation [83]. The expression of S1PR2 is correlated to hepatic fibrosis under cholestatic conditions [105]. Activation of S1PR2 has also been linked to inflammation and mitochondrial dysfunction [106,107]. Considering the high TCA in NASH patients, TCA-induced activation of S1PR2 may play an important role in promoting fibrosis. Compared to FXR and TGR5, the role of S1PR2 in metabolic disease remains largely unclear. More studies are needed to develop specific agonists or antagonists of S1PR2 as a potential therapy for NAFLD.

## 6. Conclusions and Perspectives

The pathogenesis of NAFLD is multifactorial. Recently, it has been proposed to rename NAFLD as metabolic-associated fatty liver disease (MAFLD) based on clinical evidence indicating the close association of NAFLD and metabolic syndrome [108]. The gut–liver axis plays an essential role in regulating systemic metabolism. As the major mediators of the enterohepatic circulation, BAs play critical roles in nutrition absorption and signal transduction by modulating the gut microbiome and activating different BA receptors [109]. Although FXR and TGR5 modulators have promising therapeutic potential for NAFLD, their beneficial effects are compromised by the undesired biological actions due to the ubiquitous expression of these receptors. The development of tissue- and cell-type-specific modulators of BA receptors represent a promising therapeutic strategy [89]. Identification of S1PR2 as a BA-activated receptor opened a new direction of BA research. In addition, NAFLD is not only associated with dysregulation of metabolic pathways but is also linked to dysfunction of the immune system. There is an urgent need to identify new therapeutic targets and develop tissue- or cell-type-selective agonists or antagonists for BA receptors. The combination therapy with different regimens targeting distinct and complementary mechanisms will have a more promising therapeutic potential.

## Figures and Tables

**Figure 1 cells-10-02806-f001:**
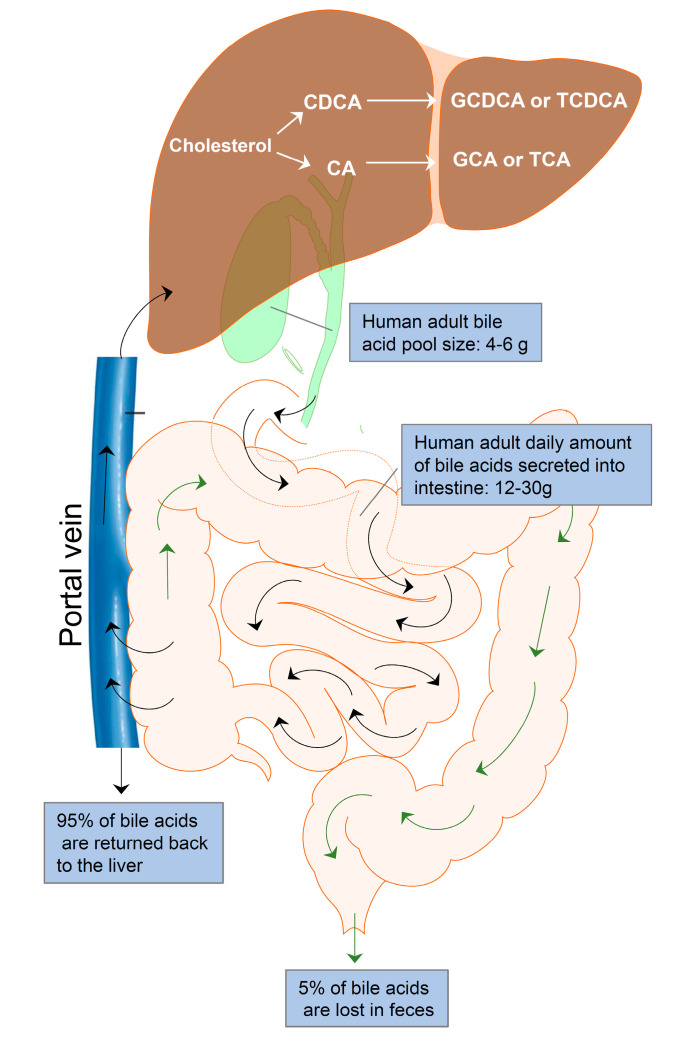
Enterohepatic bile acid circulation. Bile acids are synthesized from cholesterol in the hepatocytes to form the primary bile acids, cholic acid (CA) and chenodeoxycholic acid (CDCA). CA and CDCA are conjugated with glycine or taurine to form conjugated bile acids, GDCA, TDCA, GCA, and TCA. Bile acids are stored in the gallbladder and released into the intestine after meals for facilitating nutrition solubilization and absorption. Most bile acids (95%) are reabsorbed by ileum epithelial cells and transported to the liver via the portal vein. Only 5% of bile acids are lost in feces and replaced by *de novo* synthesis in the hepatocytes. The average adult bile pool size is about 4–6 g, and the average daily amount of bile acids secreted into the intestine is approximately 12–30 g with 3–5 circulations.

**Figure 2 cells-10-02806-f002:**
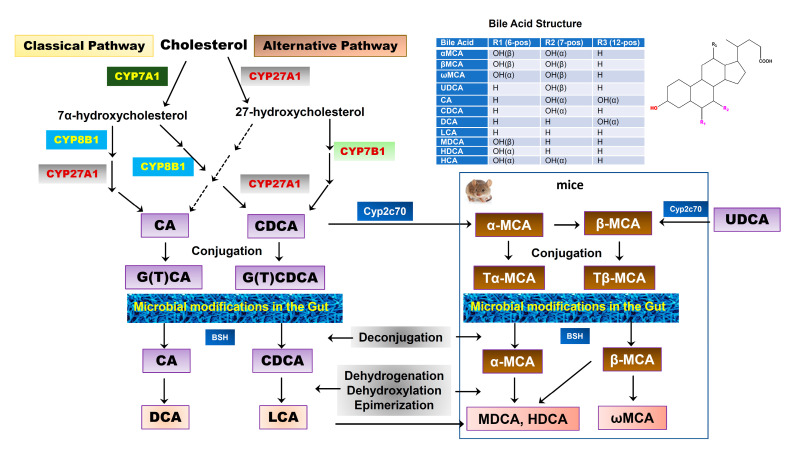
Bile acid synthesis in the liver and biotransformation in the gut. Two major pathways of BA synthesis from cholesterol have been characterized. Cholesterol-7α-hydroxylase (CYP7A1) and sterol 27-hydroxylase (CYP27A1) are the rate-limiting enzymes for the classical pathway and alternative pathway, respectively. Cholic acid (CA) and chenodeoxycholic acid (CDCA) are two primary BAs found in humans, which are conjugated with glycine or taurine. In mice, CDCA and UDCA are further converted into α-muricholic acid (α-MCA) and β-MCA by Cyp2c70, respectively. α-MCA also can be converted into β-MCA. Both α-MCA and β-MCA are conjugated with taurine. The conjugated primary BAs are deconjugated and transformed into secondary BAs by gut bacteria. In humans, CA is converted into deoxycholic acid (DCA) and CDCA is converted into lithocholic acid (LCA). In mice, α-MCA and β-MCA are converted into murideoxycholic acid (MDCA), hyodeoxycholic acid (HDCA), and ω-MCA. LCA also can be converted into MDCA and HDCA.

**Figure 3 cells-10-02806-f003:**
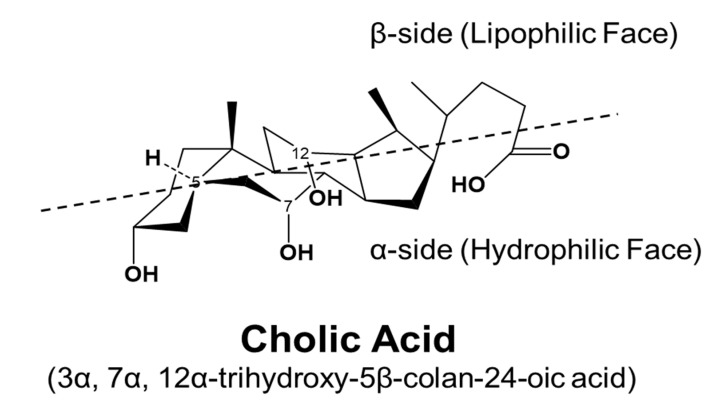
Stereostructure of cholic acid. The α-side forms a hydrophilic face, and the β-side forms a lipophilic face.

**Figure 4 cells-10-02806-f004:**
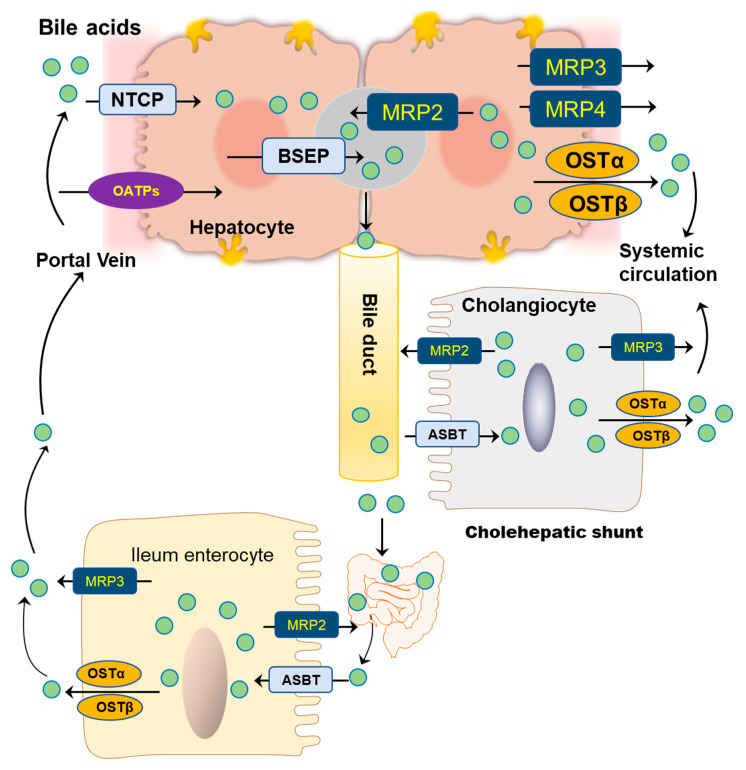
Bile acid transporters in hepatocytes, cholangiocytes, and ileum enterocytes.

**Figure 5 cells-10-02806-f005:**
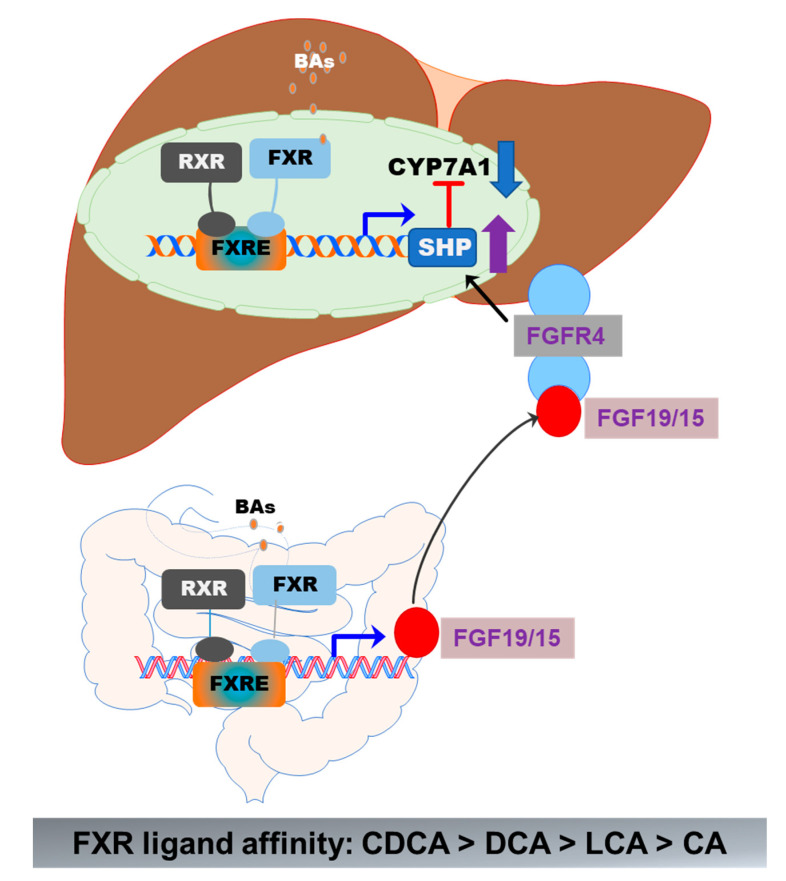
Activation of FXR in the gut–liver axis. In the liver, BA-induced activation of FXR suppresses CYP7A1 via upregulation of SHP. In the intestine epithelial cells, activation of FXR induces the release of FGF19/15, which activates FGFR4 in hepatocytes, resulting in the upregulation of SHP and downregulation of CYP7A1 expression. The formation of the FXR and RXR heterodimer further enhances the activity of FXR.

**Figure 6 cells-10-02806-f006:**
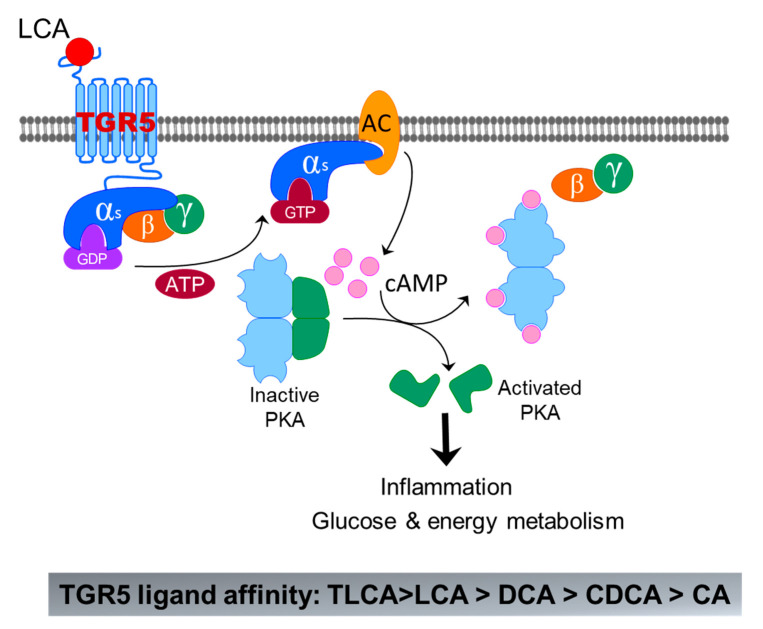
Bile-acid-induced activation of TGR5. The binding of LCA to TGR5 induces activation of Gαs, which further activates adenyl cyclase (AC) m followed by activation of protein kinase A (PKA) by cAMP. Activated PKA can activate a lot of signaling pathways related to inflammation, glucose, and energy metabolism.

**Figure 7 cells-10-02806-f007:**
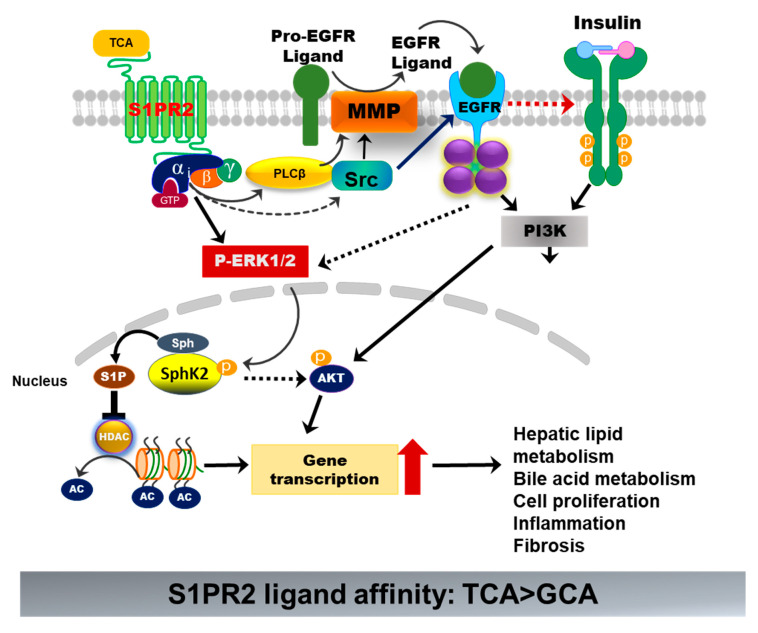
Bile-acid-induced activation of S1PR2. TCA-induced activation of S1PR2 in hepatocytes activates ERK1/2 and PI3K-Akt pathways. Activation of ERK1/2 further activates sphingosine kinase 2 (Sphk2), which generates S1P in the nucleus. S1P is a potent inhibitor of histone deacetylase (HDAC). Increased acetylation of histone promotes the transcription of key genes involved in hepatic metabolism, inflammation, cell proliferation, and fibrosis.

**Figure 8 cells-10-02806-f008:**
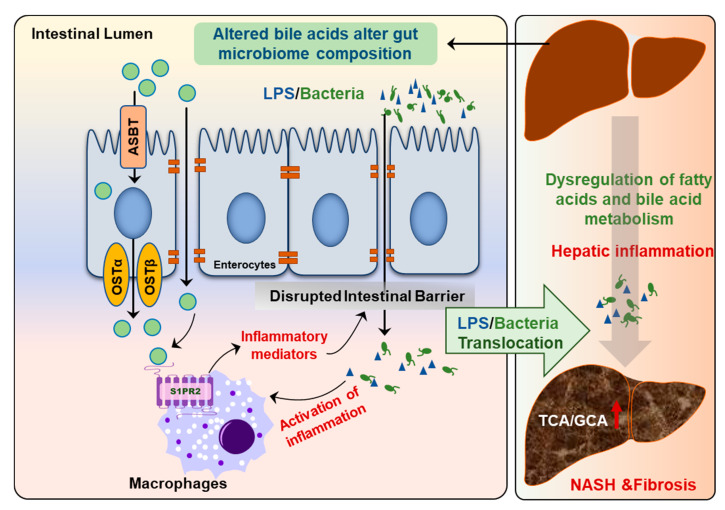
Disruption of gut–liver axis promotes NAFLD disease progression. The changes in gut BA composition and levels under metabolic stress disrupt intestinal barrier function and result in the activation of inflammatory response due to translocation of gut bacteria and bacteria-derived products, such as lipopolysaccharides (LPS). Hepatic inflammation is a major driving force to promote disease progression from NAFL to NASH.

**Figure 9 cells-10-02806-f009:**
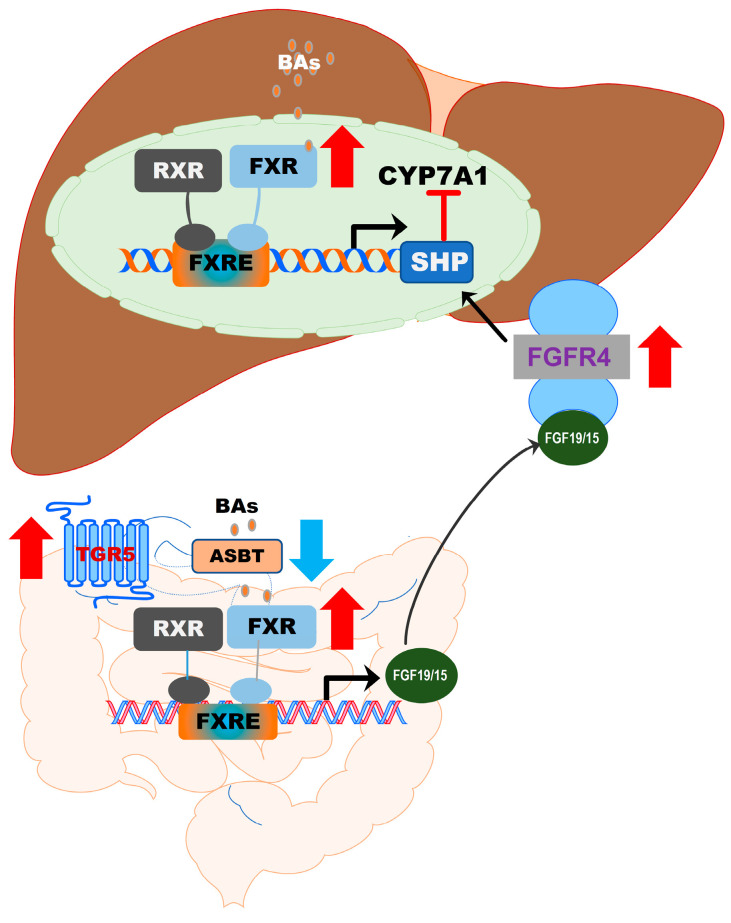
Targeting intrahepatic BA circulation represents promising therapeutics for NASH. Activation of FXR, TGR5, or inhibition of ASBT has beneficial effects on modulating hepatic lipid metabolism and inflammatory response.

**Table 1 cells-10-02806-t001:** List of the major clinical trials targeting enterohepatic BA circulation for the treatment of NASH.

Name	Targets	Mechanism	Clinical Trials	Status	Sponsor
INT-747 (Obeticholic acid)	FXR	FXR agonist	NCT03836937	Approved by FDA for PBC; Rejected by FDA for NASH	Intercept
PX-102	FXR	FXR agonist	NCT01998672	Multiple Ascending Oral Dose Phase I Study	Gilead Sciences
Tropifexor	FXR	FXR agonist	NCT02855164	Completed Phase II	Novartis
GS-9674	FXR	FXR agonist	NCT02854605	Completed Phase II	Gilead
EPY001a	FXR	FXR agonist	NCT03976687	Completed Phase I	Enyo Pharma
EPY001a	FXR	FXR agonist	NCT03812029	Phase II	Enyo Pharma
MET409	FXR	FXR agonist	NCT04702490	Phase II	Metacrine, Inc
TERN-101	FXR	FXR agonist	NCT04328077	Phase II	Terns, Inc.
Tropifexor & Licogliflozin	FXR & SGLT1/2	FXR agonist/SGLT1 inhibitor	NCT04065841	Phase II	Novartis
Tropifexor & Cenicriviroc	FXR & CCR2/5	FXR agonist/CCR2/5 antagonist	NCT03517540	Phase II	Novartis
LMB763	FXR	FXR agonist	NCT02913105	Terminated	Novartis
EDP-305	FXR	FXR agonist	NCT04378010	Phase II	Enanta Pharmaceuticals
INT767	FXR &TGR5	FXR and TGR5 dual agonist		Phase I	Intercept
Elobixibat	ASBT	ASBT inhibitor	NCT04006145	Completed Phase II	Albireo
Odevixibat	ASBT	ASBT	ASBT inhibitor	Approved for PFIC; Phase II for NASH	Albireo
NGM-282	FGFR4	FGF19 analogs	NCT04210245	Phase II	NGM Biopharmaceuticals, Inc
NGM-313	Beta-Klotho/FGF1cR	FGF19 analogs	NCT03298464	Completed Phase I	NGM Biopharmaceuticals, Inc
Cilofexor	FXR	FXR agonist	NCT03449446	Phase II	Gilead Sciences

ASBT: apical sodium-dependent bile acid transporter; CCR: C-C motif chemokine receptor; FGF: fibroblast growth factor; FGFR4: FGF receptor 4; FXR: Farnesoid X receptor; PFIC: progressive familial intrahepatic cholestasis; SGLT: sodium-dependent glucose cotransporter.

## Data Availability

Not applicable.

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
