# Peer review of "Bile Acid Receptors and the Gut–Liver Axis in Nonalcoholic Fatty Liver Disease"

_cells, 2021, doi:10.3390/cells10112806_

Round 1
Reviewer 1 Report
Most of my suggestions have been accepted.
Author Response
The manuscript has been edited by a native English speaker with extensive editing and medical research experience.
Reviewer 2 Report
I read with interest the manuscript entitled "Bile acid receptors and the gut-liver axis in nonalcoholic fatty liver disease".
Overall is a very comprehensive review describing bile acid synthesis and their biotransformation in the gut. The authors described the physiological role of bile acids by describing the receptors involved and the potential use as therapeutic targets.
Minor comments
Several studies revealed that NASH patients showed altered bile acids composition compared to non-NASH patients but results deserve more attention since differer from each other. I suggest to include in the paragraph on bile acids and NAFLD, the sites and mechanisms of insulin resistance and the link between bile acids composition and the alterations of both glucose and lipids metabolism (see tha paper by Grzych et al JHEP Rep 2020 Dec 16;3(2):100222. doi: 10.1016/j.jhepr.2020.100222).
Author Response
Response: I would like to thank this review for this critical point. I have added major findings from this paper and discussed the correlation of plasma BA levels with NASH and insulin resistance.
Reviewer 3 Report
The authors review the gut-liver axis in nonalcoholic fatty liver disease and focus on the effects of bile acid receptors. Though interesting, improvements are required for the manuscript to be published.
- The authors should add more discussions on section 5 about the potential therapeurtics of NAFLD with target agents on BA receptors for more information of readers. Also have illustrations for a clear picture.
- Table 1 was lacking for the current trials on the target of BA receptors.
- There is no caption of figure 4.
Author Response
The authors review the gut-liver axis in nonalcoholic fatty liver disease and focus on the effects of bile acid receptors. Though interesting, improvements are required for the manuscript to be published.
- The authors should add more discussions on section 5 about the potential therapeurtics of NAFLD with target agents on BA receptors for more information of readers. Also have illustrations for a clear picture.
Response: I agree with this reviewer. We included the current clinical trials with FXR and TGR5 agonists for NAFLD/NASH. In addition, new Figure 9 illustrates the major targets of the clinical trials for NASH.
- Table 1 was lacking for the current trials on the target of BA receptors.
Response: Most of the current trials on the target of BA receptors are for PSC or PBC. I added one for NASH (NCT03449446) in Table 1.
- There is no caption of figure 4.
Response: The detailed description of Figure 4 is present in the main text.
Round 2
Reviewer 3 Report
The author responded and modified the manuscript to reviewers' suggestion. I think it is good for publication in the Journal.